# Energy-Saving Control of Hybrid Tractors Based on Instantaneous Optimization

Junjiang Zhang [1,2,3], Ganghui Feng [1], Liyou Xu [1,3,*], Xianghai Yan [1,2,3], Wei Wang [4] and Mengnan Liu [2,4]

1  College of Vehicle and Traffic Engineering, Henan University of Science and Technology, Luoyang 471003, China
2  State Key Laboratory of Power System of Tractor, Luoyang 471039, China
3  Henan Province Collaborative Innovation Center for Advanced Manufacturing of Mechanical Equipment, Luoyang 471003, China
4  YTO Group Corporation, Luoyang 471004, China
*  Correspondence: xlyou@haust.edu.cn; Tel.: +86-136-6387-3262

**Abstract:** In this study, an energy-saving control strategy based on instantaneous optimization is proposed to improve the energy efficiency of hybrid tractors. Using a parallel diesel–electric hybrid tractor as the research object, the topological and working characteristics were analyzed, and a coupled dynamic model of rotary tillage and tractor plow was constructed. Aiming to minimize the equivalent fuel consumption of the entire machine, the motor and diesel engine torques were taken as the control variables, and the state of charge of the power battery was taken as the state variable. Subsequently, an energy-saving control strategy based on instantaneous optimization is proposed. Finally, a simulation experiment was carried out using MATLAB to verify the effectiveness of the energy-saving control strategy based on instantaneous optimization. Compared with the energy-saving control strategy based on power-following, the results show that energy-saving control strategy based on instantaneous optimization can reasonably control the operating state of the diesel engine and motor. Therefore, the diesel engine and motor work in the high-efficiency area, and effectively reduce the equivalent fuel consumption of the tractor during field operation. Under rotary tillage and plowing conditions, equivalent fuel consumption is reduced by 4.70% and 6.31%, respectively.

**Keywords:** hybrid tractor; instantaneous optimization; energy-saving control; equivalent fuel consumption

## 1. Introduction

Recently, with the introduction of various agricultural machinery subsidy policies, there has been an increase in the use of tractors as the main power machinery for agricultural production operations. The national "14th Five-Year Plan and 2035 Long-term Goals" proposed goals regarding a carbon emission peak in 2030 and carbon neutrality in 2060. At this stage, agricultural machinery mainly uses diesel engines as the main power source, which consume a significant amount of fuel during operation and result in greenhouse gas emissions [1]. Under the double pressure of the global energy crisis and environmental pollution, it is particularly important to design and develop energy-saving and environmentally friendly agricultural machinery vehicles [2–4]. With the improvements in hybrid vehicle technology, hybrid tractors have been developed [5]. Hybrid tractors are energy-saving and environmentally friendly agricultural machinery vehicles that have the advantages of both traditionally fueled tractors and electric tractors [6–8]. The energy conversion of a series hybrid tractor is repeated during operation, which results in low energy utilization efficiency. Meanwhile, parallel hybrid tractors can be powered directly by diesel engines or electric motors. It has no secondary energy conversion and realizes high energy utilization efficiency. Therefore, based on the topology of a parallel diesel-electric

hybrid tractor, this study develops an energy-saving control strategy to improve the energy utilization efficiency of the entire machinery.

Using the energy-saving control strategy as the core control strategy of a hybrid tractor directly affects the performance of a hybrid tractor. Currently, energy-saving control strategies can be mainly divided into two categories: rule-based control strategies and optimization-based control strategies [9,10].

Rule-based control strategies are inexpensive to develop and simple to implement, and they are widely used in various types of hybrid vehicles. Luo et al. [11] proposed a fuzzy reasoning energy management strategy for a series of diesel-electric hybrid tractors. The engine demand power is determined according to the self-set fuzzy reasoning rule table. The corresponding results demonstrate that the fuel economy is improved by 20.92% compared with the power following control strategy. Xu et al. [12] proposed a predefined energy management strategy for extended-range electric tractors; they indicated that the fuel consumption was reduced by 34.22% in the continuous transition operation mode. Based on series hybrid tractors, Fang et al. [13] proposed a fuzzy control energy management strategy that differentiated between electric vehicles and electric tractors. Their results showed that the battery of the state of charge (SOC) curve shows the slowest change when the fuzzy control energy management strategy is adopted. However, rule-based control strategies are certain and according to the designer's experience, they do not exhibit good adaptability to working conditions [14].

Optimization-based control strategies are solved by minimizing or maximizing a cost function, which is generally a measure of the control target. Lee et al. [15] established a power shunt ratio strategy based on a drivetrain simulation model. The power allocation strategy of a hybrid electric tractor was optimized using a deterministic dynamic programming algorithm. Simulation results show that the proposed control strategy can reduce the fuel consumption of a hybrid electric tractor. Spano et al. [16] proposed a multi-objective particle swarm optimization algorithm to determine the optimal power system layout of parallel P2 hybrid electric vehicles, aiming to maximize fuel economy and minimize production costs. The results showed that the control algorithm can improve the fuel economy of HEV and reduce HEV production costs. Qian et al. [17] proposed a calculation method based on the fuzzy PID torque recognition coefficient K. Subsequently, they used the particle swarm-ant colony combination optimization algorithm to optimize the key control parameters in the control strategy. Their results indicated that the fuel consumption and emissions are reduced by ensuring the dynamics of the entire vehicle. However, the solution of deterministic dynamic programming algorithms requires extracting control rules, which is computationally intensive and time-consuming. The particle swarm optimization algorithm has strong global search ability and is a simple algorithm, but it has poor local search ability. Meanwhile, the control strategy of particle swarm-ant colony combination optimization algorithm is more complicated. Recently, energy-saving control strategies based on instantaneous optimization have become a research hotspot in the vehicle control field, owing to their advantages of fast calculation speed, good control effect, smaller calculation amount than dynamic programming algorithms, and simple control algorithms.

In this study, a diesel-electric parallel hybrid tractor is considered as the research object. Meanwhile, an energy-saving control strategy based on instantaneous optimization of the required torque is proposed [18–23]. By optimizing the torque of both the diesel engine and motor, the required torque of the entire machine can be distributed in real time, which reduces the equivalent fuel consumption of the entire machine while ensuring power stability. First, the topology and performance parameters of the hybrid tractor are introduced; then, the main components are simulated and modeled [24,25]. Based on the entire machine model, an energy-saving control strategy based on instantaneous optimization was designed. The simulation results were analyzed and compared with the power-following energy-saving control strategy. Finally, the conclusions of this study are presented.

## 2. Tractor Topology and Main Parameters

### 2.1. Subsection Model of Hybrid Tractor Drivetraino

Figure 1 shows the topology of a parallel diesel–electric hybrid tractor, which has two drive systems, namely a diesel engine and an electric motor. The diesel engine is the main power source, and the electric motor is the auxiliary power source [26]. The engine and motor output torques are transmitted to the transmission input shaft through the torque coupler, followed by the transmission output power. It is used as the power input of the central transmission device and power take-off (PTO) shaft. The vehicle controller and power battery, low-voltage battery, diesel engine, clutch, drive motor, transmission, AC/DC module, and DC/DC module are connected through the CAN bus. According to the total power demand of the entire machine and SOC value of the power battery, the torques of the diesel engine and motor are dynamically distributed according to the control strategy and algorithm of the entire machine. Accordingly, the tractor can achieve the best power and economic performance.

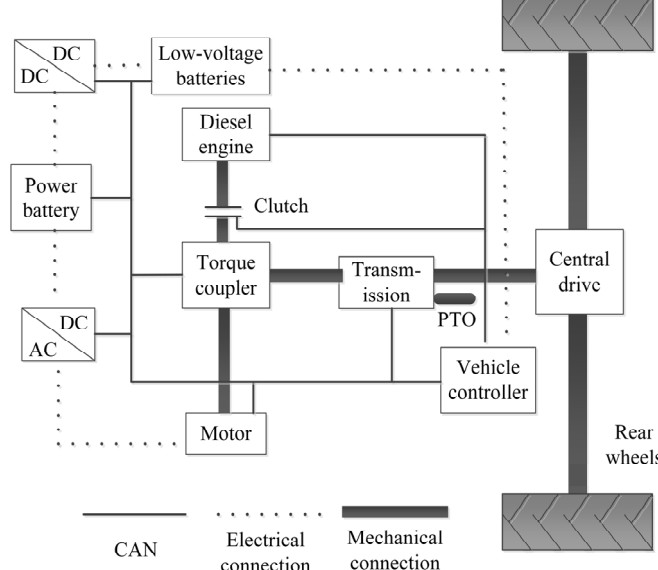

**Figure 1.** Topology of a diesel–electric parallel hybrid tractor.

### 2.2. Subsection Model of Hybrid Tractor Drivetraino

In this study, the energy-saving control strategy of a 220 hp hybrid tractors is investigated. The main components of the hybrid tractor were selected according to its working conditions [27]. The specific parameters are listed in Table 1.

**Table 1.** Topology of a diesel–electric parallel hybrid tractor.

| Name | Parameter | Value (Unit) |
|---|---|---|
| Diesel engine | Rated power | 162 (kW) |
| | Rated speed | 2500 (rpm) |
| | Maximum torque | 800 (Nm) |
| Motor | Rated power | 30 (kW) |
| | Rated speed | 3000 (rpm) |
| | Rated torque | 96 (Nm) |
| Power battery | Energy capacity | 70 (Ah) |
| | Rated voltage | 360 (V) |
| | SOC | 0.90–0.25 |

### 2.3. Determination of Theoretical Speed and Transmission Ratio

The design of the transmission gear ratio is based on the working characteristics of the tractor, speed range of various working conditions of the current tractor, and the output characteristics of the hybrid tractor coupling system. Subsequently, the main reducer and gearbox transmission ratio, and its theoretical speeds are matched and calculated [28,29].

When rotary tillage is performed, the traveling speed of the tractor is 4–5 km/h, and the theoretical value of the PTO speed is 540 rpm. From this, the speed ratio of the main reducer is determined as $i_0$, which is taken as 19.10. Seven forward gears are designed, including three transport gears, two working gears, and two amble gears [30,31]. The corresponding transmission ratio increases sequentially, while the theoretical speed decreases sequentially. The specific parameters used are listed in Table 2.

**Table 2.** Hybrid tractor transmission ratios and theoretical speeds.

| Forward Gear | Transport III | Transport II | Transport I | Working II | Working I | Amble II | Amble I |
|---|---|---|---|---|---|---|---|
| Ratio | 0.864 | 1.377 | 2.307 | 3.296 | 4.963 | 7.405 | 11.208 |
| Theoretical speed (km/h) | 39.980 | 28.668 | 17.164 | 11.977 | 7.654 | 5.231 | 3.522 |

## 3. Hybrid Tractor Model Building

Based on the topology of the hybrid tractor, its main components were modelled, including the transmission system, rotary tillage unit dynamic, plowing unit dynamic, tire, motor, diesel engine, power battery model. Subsequently, the simulation model of the machine was built.

### 3.1. Model of Hybrid Tractor Drivetraino

The required torque of the hybrid tractors provided by the motor and diesel engine. The required torque of the entire machine is obtained at the input end of the torque coupler, which can be expressed as follows:

$$T_{req} = (T_m \cdot \eta_m + T_e \cdot \eta_e) \tag{1}$$

where $T_m$ and $T_e$ are the motor and diesel engine torques, respectively. $\eta_m$ and $\eta_e$ represent the working efficiencies of the motor and diesel engine, respectively. $T_{req}$ is the torque required for the torque coupler input.

According to the tractor working speed and parameters of each component used to calculate the power source speed, this study considers the diesel engine speed as the power source speed, which can be expressed as follows:

$$n_{tire} = \frac{v}{0.377 \cdot r} \tag{2}$$

$$n_e = n_{tire} \cdot i_t \cdot i_0 \tag{3}$$

where $i_t$ and $i_0$ are the transmission and main reducer speed ratios, respectively. $n_{tire}$ and $n_e$ are the drive and diesel engine speeds, respectively. $v$ is the speed of the hybrid tractor during operation. $r$ is the driving wheel radius of the hybrid tractor.

### 3.2. Dyanmic Model of Rotary Tillage Unit

The power balance relationship characterizing the working time group of the hybrid tractor traction rotary cultivator is formulated as follows:

$$P_{req}(T_{req}, n_{req}) = \left( \frac{P_{drive}}{\eta_{zj}\eta_b} + \frac{P_r}{\eta_b} \right) / \eta_o \tag{4}$$

$$P_{drive} = \frac{\left( mgf \cos \alpha + mg \sin \alpha + m\delta\dot{v} + \frac{C_d A v^2}{21.15} \right) v}{3600} \tag{5}$$

$$P_r = \frac{P_c + P_{th} + P_a + P_h}{\eta_r} \tag{6}$$

where $P_{drive}$ and $P_r$ are the tractor travel power and rotary cultivator power consumptions, respectively. $\eta_r$, $\eta_{zj}$, $\eta_b$, and $\eta_o$ represent the rotary tillage unit mechanical transmission efficiency, main reducer transmission efficiency, transmission efficiency, and torque-coupler efficiency, respectively. $P_c$, $P_{th}$, $P_a$, and $P_h$ are the cutting power consumption, throwing earth power consumption, rotary cultivator forward power consumption and power required to overcome the soil horizontal reaction forces, respectively. $m$ is the tractor mass. $f$ is the rolling resistance coefficient. $\delta$ is the mass conversion of the factor. $\alpha$ is the tilt of the ground. $C_d$ and $A$ are the tractor drag coefficient and windward area, respectively.

When the tractor is operating at a low speed, the influence of air resistance and acceleration resistance on the tractor can be ignored [28].

The hybrid tractor is equipped with a double-acting clutch, which can realize independent control of the PTO power [29]. The relationship formula of the rotary tillage operation timing group is as follows:

$$P_r = P_{PTO} = \frac{n_{PTO} T_{PTO}}{9550} \tag{7}$$

$$v = v_r \tag{8}$$

where $P_{PTO}$, $T_{PTO}$, and $n_{PTO}$ are the power, torque, and speed of PTO, respectively. $v_r$ is the forward speed of the rotary cultivator.

### 3.3. Dyanmic Model of Plowing Unit

When the tractor is working, its driving force must overcome the rolling resistance and other driving resistances before it can be operated. The balance between the driving force $F_{TN}$ and various resistances when the tractor is operating is formulated as follows:

$$F_{TN} = F_g + F_f + F_p + F_{Af} + F_i \tag{9}$$

where $F_g$, $F_f$, $F_p$, $F_{Af}$, and $F_i$ are the tillage, rolling, slope, air, and acceleration resistances, respectively. $F_{TN}$ is the driving force.

When the tractor is operating at a low speed, the influence of air and acceleration resistances on the tractor can be ignored.

Under normal circumstances, the tractor drive force $F_{TN}$ is primarily determined by the tillage resistance $F_g$ when the supporting agricultural tools are working. The calculation formula is stated as follows:

$$F_g = Z \cdot b_l \cdot h_k \cdot k \tag{10}$$

where $Z$ denotes the number of plowshares. $b_l$ and $h_k$ represent the individual plow width and depth, respectively. $k$ is the soil specific resistance coefficient.

The power demand at the input end of the torque coupler when the hybrid tractor pulls the plow unit can be expressed as follows:

$$P_{req}(T_{req}, n_{req}) = \frac{F_{TN} v}{\eta_{zj} \eta_b \eta_o} \tag{11}$$

### 3.4. Tire Model

The Duggof model belongs to the theoretical model and is suitable for the studying vehicle dynamics control algorithms. Therefore, the Duggof tire model is used to calculate the driving force of the driving wheel [32], which can be expressed as follows:

$$F_q = \begin{cases} F_Z\left[\varphi - \varphi^2 \frac{F_Z(1-\zeta)}{4c\zeta}\right], & \frac{c\zeta}{1-\zeta} \geq \frac{\varphi F_Z}{2} \\ \frac{c\zeta}{1-\zeta}, & \frac{c\zeta}{1-\zeta} \leq \frac{\varphi F_Z}{2} \end{cases} \tag{12}$$

where $F_q$ and $F_z$ are the driving and loading forces of the drive wheel, respectively. $\varphi$ and $\zeta$ represent the slip rate of the corresponding drive wheel and adhesion factor of the corresponding drive wheel, respectively. $c$ is the horizontal distance of the hitch traction point from the center of the rear wheel.

### 3.5. Motor Model

A permanent magnet synchronous motor with superior performance is selected as the electric drive system of the tractor, which can operate not only in the forward direction, but also in the reverse direction. Moreover, it has the working characteristics of low-speed constant torque and high-speed constant power. The formula characterizing the relationship between the power, speed, and torque is stated as follows:

$$P_m = \frac{n_m \cdot T_m}{9550} \tag{13}$$

where $n_m$ is the speed of the motor.

The motor model is established via a numerical model method, and the relationship between motor system efficiency, torque, and speed are obtained via the spline interpolation method based on experimental motor efficiency experimental data. The relationship is determined and unique, which is suitable for the studied control strategies. The numerical model of motor efficiency is shown in Figure 2.

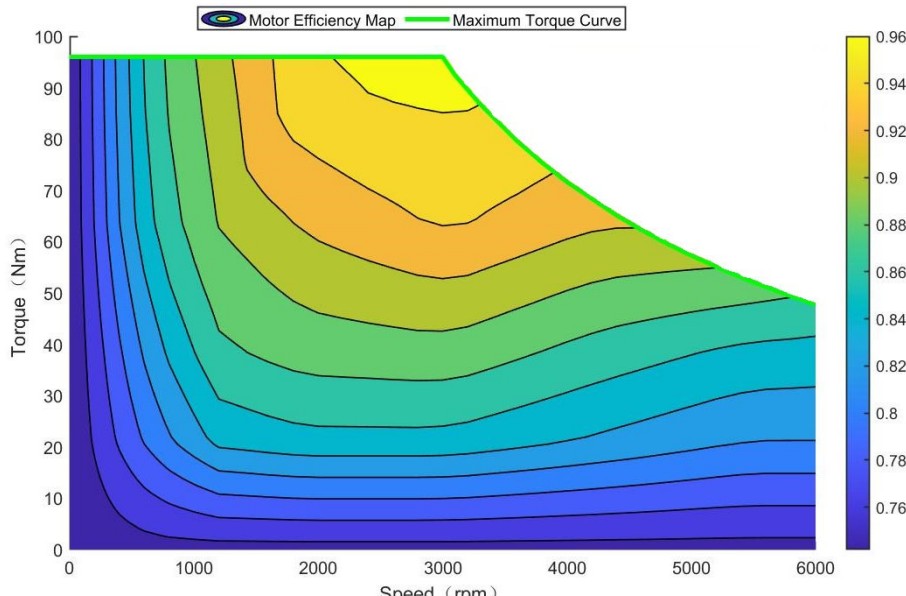

**Figure 2.** Motor model MAP diagram.

### 3.6. Diesel Engine Model

The diesel engine of the corresponding power is selected according to the use conditions of the hybrid tractor considering only the input and output parameters of the engine. The formula characterizing their relationship is stated as follows:

$$P_e = \frac{n_e \cdot T_e}{9550} \tag{14}$$

where $n_e$ denotes the speed of the diesel engine.

Diesel engine modeling methods are mainly divided into two types: theoretical modeling and numerical modeling methods. The theoretical modeling method is based on the structural parameters of the engine, using thermodynamic theory, combustion theory, fluid mechanics, and heat transfer theory to establish a mathematical model of the engine working process. The numerical modeling method tests the load characteristics and speed characteristic curves of the engine by building an experimental bench of the engine and then constructing a numerical model by interpolation fitting. This study mainly uses the engine output characteristics to study the drive system of the whole machine, and only considers the relationship between the input and output parameters of the engine. Therefore, the method of measured modeling is adopted. On the basis of the engine steady-state test data, the number table or formula is used to fit to obtain an accurate and simple engine numerical model [33], as shown in Figure 3.

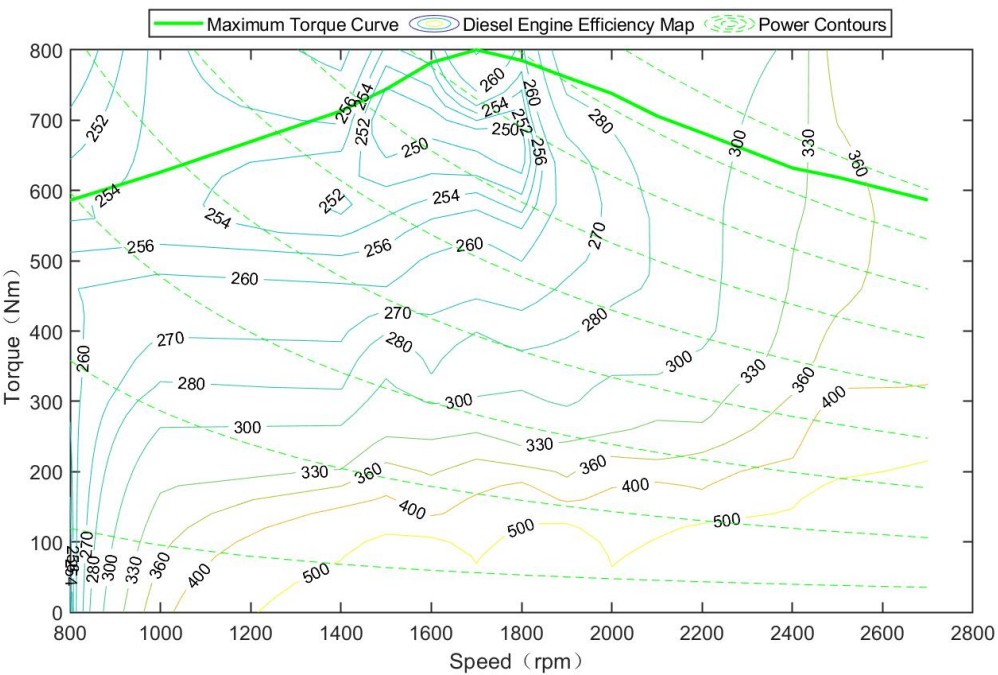

**Figure 3.** Diesel engine model MAP diagram.

### 3.7. Power Battery Model

The power battery model describes the external characteristics of the power battery during operation, and most of the equivalent circuit models are currently used. Because the model has good applicability to various working states of power batteries, the equation of state of the model can be derived.

Therefore, the equivalent internal resistance model in the equivalent circuit model is used here. The power battery is equivalent to an ideal voltage source and a circuit model with a resistor connected in series. The mathematical equations are simple and easy to calculate and model, as shown in Figure 4.

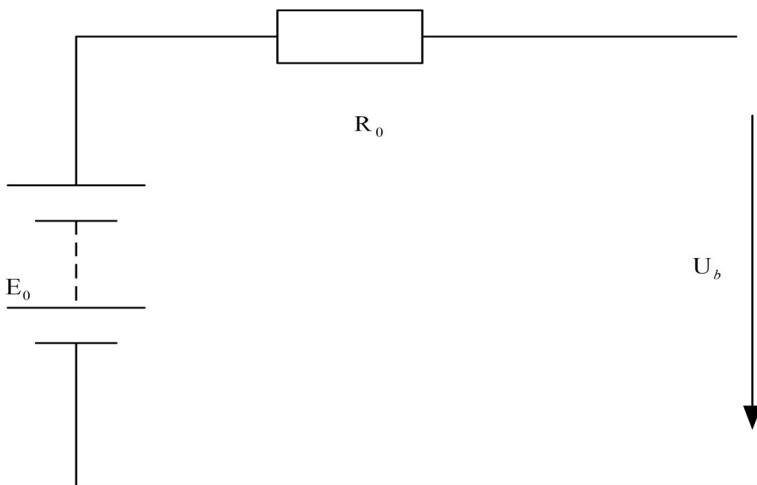

**Figure 4.** Power battery internal resistance model.

According to Ohm's law, the voltage characteristic equation of a power battery is stated as follows:

$$U_b = E_0 - I_b R_0 \tag{15}$$

where $U_b$ and $E_0$ are the power battery output voltage and terminal voltage, respectively. $I_b$ and $R_0$ represent the output current and internal resistance of the power battery, respectively.

Ignoring the influence of the internal resistance and discharge factors of the power battery on the electromotive force $E_0$ and setting it as a constant, the output power equation of the power battery can be stated as follows:

$$P_{b\max} = \frac{U_b I_b}{1000} = \frac{(E_0 - I_b R_0) I_b}{1000} \tag{16}$$

The charge and discharge power $P_{bat}(t)$ of the battery is positive when discharged and negative when charging. This parameter is determined as follows:

$$P_{bat} = \frac{P_m}{\eta_{bat}} \tag{17}$$

where $\eta_{bat}$ is the battery charging and discharging efficiency.

The ampere-hour integral method is used to calculate the change in the SOC value of the power battery, which is formulated as follows:

$$SOC(t) = SOC_0 - \frac{\int_0^t I_b(t)dt}{Q_b} \tag{18}$$

where $Q_b$ denotes the rated power battery capacity. $SOC_0$ represents the initial the state of charge value.

### 3.8. Power Battery Model

Based on the characteristics of the hybrid tractor transmission system, the simulation model of the entire machine is built using MATLAB. A simplified diagram of the entire machine model is shown in Figure 5. The simulation model includes the dynamic model of the unit (rotary tillage and plowing), motor model, diesel engine model, transmission system model, battery model, and tire model. $F_{TN}$ and $v$ are the resistance and travel speed of the tractor, respectively, which are determined based on the operating conditions of the tractor and are the output parameters of the dynamic model of the unit. Drivetrain model

output torque and speed ($T_{req}$ and $n_{req}$) at the input end of the torque coupler. The motor and diesel engine models receive the instructions ($T_{mreq}$, $n_{mreq}$, $T_{ereq}$, and $n_{ereq}$).

$$\begin{cases} T_{mreq} = T_m \cdot \eta_m \\ T_{ereq} = T_e \cdot \eta_e \end{cases} \tag{19}$$

where $T_{mreq}$ and $T_{ereq}$ are the torques required for the motor and diesel engine, respectively, at the input of the torque coupler.

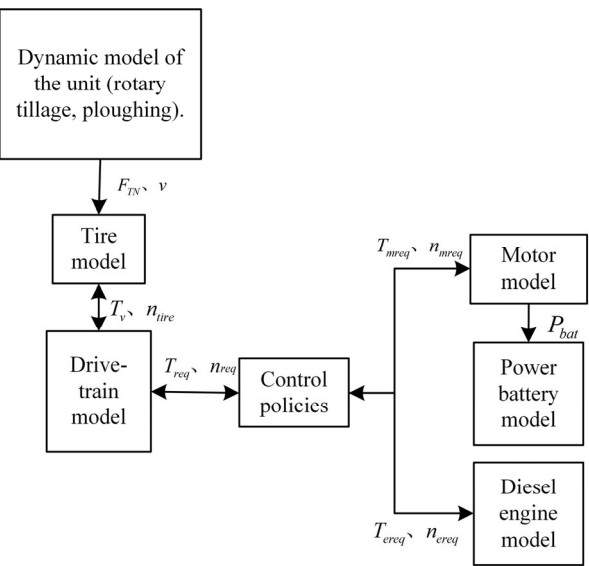

**Figure 5.** Schematic diagram of the whole machine simulation model.

Finally, the diesel engine and motor work according to the command and output the corresponding torque and speed. Through the drivetrain model, the power is transmitted to the tire model to ensure the normal operation of the whole machine.

## 4. Energy Saving Control Strategy Design

Based on the entire machine model, two energy-saving control strategies (based on instantaneous optimization and power follower) were designed. The design process of an energy-saving control strategy based on instantaneous optimization is introduced in detail, while that of a comparative control strategy (power-following) is briefly introduced.

### 4.1. Energy-Saving Control Strategy Based on Instantaneous Optimization

First, according to the topological structure and main component model of hybrid electric tractor, an energy-saving control optimization model of the entire machine is designed. Then, the optimization solution is carried out according to the instantaneous optimization algorithm. Finally, the solution flow is discussed in detail.

#### 4.1.1. Optimization Model of Energy-Saving Control Strategy

A hybrid tractor has two energy sources, electric power and fuel. To unify the energy, the equivalent fuel consumption is used for the evaluation. The goal of energy management is to minimize the equivalent fuel consumption by optimizing and rationalizing the operating state between the diesel engine and motor. The equivalent fuel consumption in the operation process of the hybrid electric tractor, namely, the objective function, can be expressed as follows:

$$Q_c(t) = \int_0^t f\left( Q_f(T_e, n_e) + \frac{j_m P_m(T_m, n_m)}{j_e \eta_{bat} \eta_m} \right) dt \tag{20}$$

$$Q_f(t) = \frac{f_e \cdot P_e(T_e, n_e)}{1000 \cdot 3600 \cdot 0.84} \tag{21}$$

where $Q_c(t)$ and $Q_f(t)$ are the equivalent fuel consumption and instantaneous fuel consumption, respectively. $t_f$ represents the terminal moment. $j_e$ and $j_m$ are the prices per liter of oil and per kWh of electricity, respectively. $f_e$ is engine fuel consumption at that moment.

According to the calculation of the battery SOC value using Equation (18), the system state equation can be obtained as follows:

$$S\dot{O}C(t) = -\frac{I_b(t)}{Q_b} = -\frac{U_b(t) - \sqrt{U_b^2 - 4P_b(t)R_0(t)}}{2R_0(t)Q_b} \tag{22}$$

The control variables of the system are $T_e(t)$ of the diesel engine and $T_m(t)$ of the motor torque. The relationship between them and the required torque is introduced based on Equation (1).

Because the working capacity of each component is limited by realistic conditions, the system must satisfy the following constraints:

$$\begin{cases} T_{mmin}(n_{m,t}) \leq T_m(t) \leq T_{mmax}(n_{m,t}) \\ T_{emin}(n_{e,t}) \leq T_e(t) \leq T_{emax}(n_{e,t}) \\ SOC_{min} \leq SOC(t) \leq SOC_{max} \end{cases} \tag{23}$$

where $T_{mmin}$ and $T_{mmax}$ are the minimum and maximum torques of the motor, respectively. $T_{emin}$ and $T_{emax}$ are the minimum and maximum torque of the diesel engine, respectively. $SOC_{min}$ and $SOC_{max}$ represent the minimum and maximum values allowed by the SOC value of the power battery, respectively.

Equation (22) constitutes the permissive reach of the control variables.

### 4.1.2. Establish the Optimal Torque Distribution Table

To optimize the energy-saving control, an instantaneous optimization control strategy is adopted to solve this problem. The SOC value of the power battery is taken as the state variable. Moreover, the diesel engine torque $T_e$ and motor torque $T_m$ are taken as the control variables to address the optimal torque distribution table. The specific process is illustrated in Figure 6.

1.  According to the typical working conditions of a tractor, a set of operating parameters (speed ratio $i_t$, required torque $T_{req}$, and power source speed $n_e$) within a short period of time are used as the system input parameters;

2.  In the value range, step sizes $\Delta T_{req}$ and $\Delta n_e$ are used to discretize the required torque and power source speed, respectively;

$$\begin{cases} k = 0 : \Delta T_{req}(t) : T_{req}(t) \\ j = 0 : \Delta n_e(t) : n_e(t) \end{cases} \tag{24}$$

3.  According to the speed $n_e$, determine the maximum torques $T_{emax}$ and $T_{mmax}$ that the diesel engine and motor can achieve at this speed;

4.  Take the SOC state value of the power battery as the state variable, the torques of motor and diesel engine as the control variable, and minimum equivalent fuel consumption as the objective function $Q_c$, to determines the optimal instantaneous torque of the diesel engine and motor;

5.  Record the torque of the diesel engine and motor corresponding to the required instantaneous torque and speed until the end of $t_f$ at the final moment, summarize the data at all moments, and form the optimal torque distribution table.

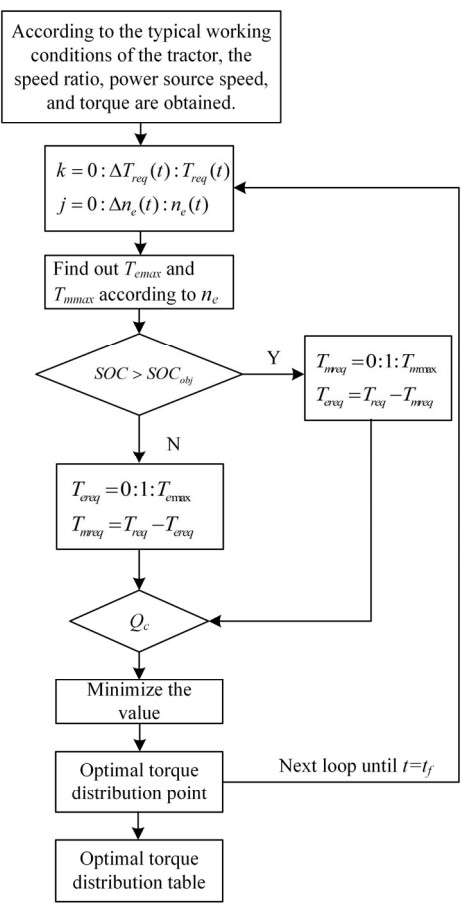

**Figure 6.** The flowchart of optimal torque distribution table solves.

According to the solution process presented in Figure 6, the torque distribution MAP of the diesel engine and motor can be obtained under at given speed and torque setpoint of the input of the torque coupler. This is illustrated in Figure 7.

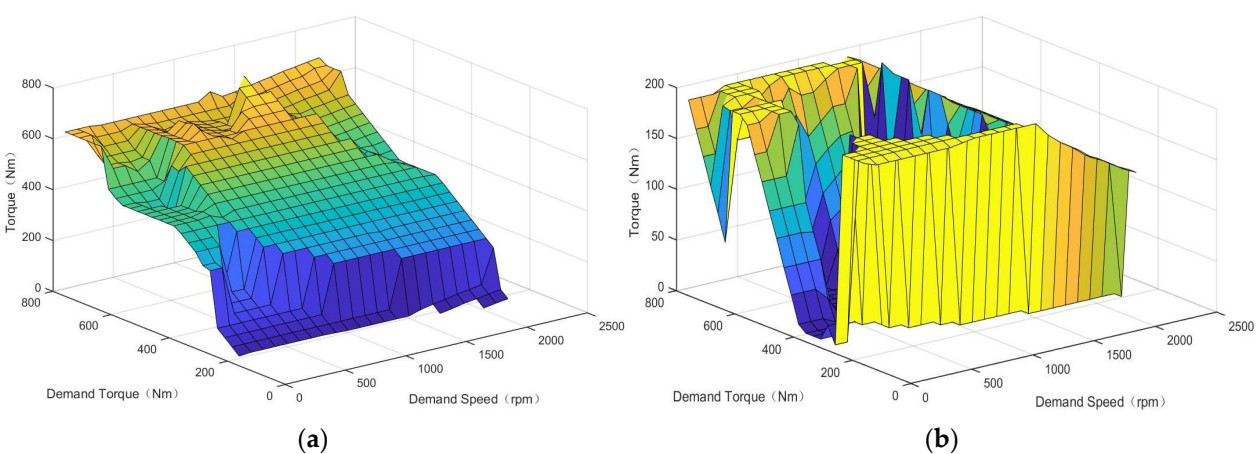

**Figure 7.** Torque distribution under hybrid drive. (**a**) Torque of the diesel engines. (**b**) Torque of the motor.

### 4.1.3. Instantaneous Optimization Control

According to the instantaneous optimization control strategy, the optimal torque distribution of the diesel engine and motor during tractor operation is solved, as shown in Figure 8.

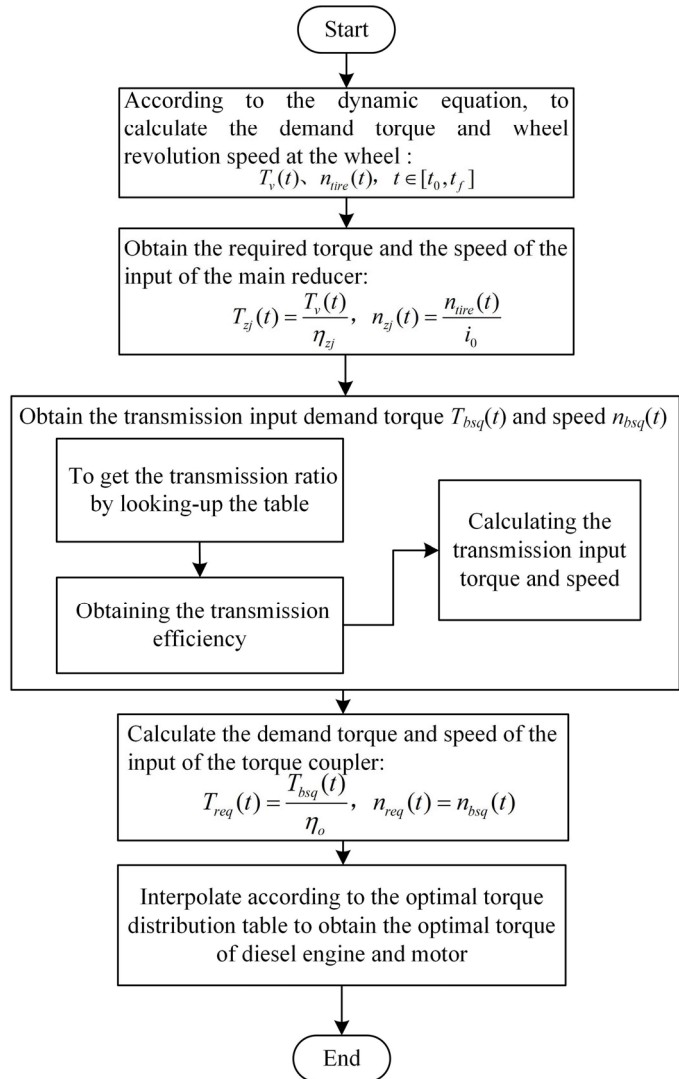

**Figure 8.** Torque distribution flowchart based on instantaneous optimization.

The specific solving steps are shown as follows:

1. According to the dynamic equation, calculate the required torque $T_v(t)$ and wheel speed $n_{tire}(t)$ at the wheel of the entire machine.

$$n_{tire}(t) = \frac{v(t)}{0.377 \cdot r} \tag{25}$$

$$T_v(t) = \frac{9550 \cdot P_v(t)}{n_{tire}(t)} \tag{26}$$

where $P_v(t)$ is the required power at the wheels of the entire machine.

2. Obtain the required torque $T_{zj}(t)$ and speed $n_{zj}(t)$ for the input of the main reducer.

$$\begin{cases} n_{zj}(t) = \frac{n_v(t)}{i_0} \\ T_{zj}(t) = \frac{T_v(t) \cdot i_0}{\eta_{zj}} \end{cases} \tag{27}$$

3.  Obtain the required transmission input torque $T_{bsq}(t)$ and speed $n_{bsq}(t)$. First, according to the torque required by the main reducer and the tractor speed, the transmission ratio is calculated by looking up the table. Then, the transmission efficiency is obtained by looking up the transmission ratio and torque table. Finally, the transmission input torque and speed are calculated using the transmission speed ratio and efficiency.

$$\begin{cases} n_{bsq}(t) = \frac{n_{zj}(t)}{i_t} \\ T_{bsq}(t) = \frac{T_{zj}(t)}{i_t \cdot \eta_b} \end{cases} \tag{28}$$

4.  Calculate the required torque $T_{req}(t)$ and speed $n_{req}(t)$ of the torque coupler input.

$$\begin{cases} n_{req}(t) = n_{bsq}(t) \\ T_{req}(t) = \frac{T_{bsq}(t)}{\eta_o} \end{cases} \tag{29}$$

5.  The required torque $T_{req}(t)$ and speed $n_{req}(t)$ at the input end of the torque coupler obtained in the previous step are interpolated according to the optimal torque distribution table. Obtain the optimal torque distribution of the diesel engine and motor during tractor operation.

### 4.2. Energy-Saving Control Strategy Based on Power Following

4.2.1. Control Principle Based on Power Following

As a comparative control strategy, the power following energy-saving control strategy, which is also a rule-based control strategy, is considered. Taking the ratio of the rated power between the diesel engine and motor as the distribution ratio, the power demand of the entire machine is allocated according to the fixed proportion to fully leverage the working capacity of the diesel engine and motor.

4.2.2. Solving Process of Power Following Energy-Saving Control

According to the operating conditions of the hybrid electric tractor and dynamic model analysis conducted on the entire machine, the power required of the entire machine $P_{req}$ and diesel engine speed $n_e$ can be obtained. Accordingly, the required torque of the entire machine $T_{req}$ can be obtained.

According to the rated power of the diesel engine and motor, the torque distribution proportion coefficient K can be determined, as follows:

$$K = \frac{P_{erated}}{P_{erated} + P_{mrated}} \tag{30}$$

where $P_{erated}$ and $P_{mrated}$ are the rated powers of the diesel engine and motor, respectively.

According to the torque distribution proportion coefficient K, the respective working torques of the diesel engine and motor can be obtained using the following equations:

$$\begin{cases} T_{ereq} = K \cdot T_{req} \\ T_{mreq} = (1 - K) \cdot T_{req} \end{cases} \tag{31}$$

The constraint conditions are as follows:

$$\begin{cases} T_{mmin}(n_{m,t}) \le T_m(t) \le T_{mmax}(n_{m,t}) \\ T_{emin}(n_{e,t}) \le T_e(t) \le T_{emax}(n_{e,t}) \\ SOC_{min} \le SOC(t) \le SOC_{max} \end{cases} \tag{32}$$

## 5. Result Analysis

Two types of energy-saving control strategies were simulated and tested for the two typical working conditions of rotary tillage and plowing. Accordingly, the simulation results were compared and analyzed. The superiority of the energy-saving control strategy based on instantaneous optimization was verified.

### 5.1. Analysis of Results Obtained Rotary Tillage Condition

During the rotary tillage operation of the tractor, the PTO works independently and is unaffected by the driving conditions of the tractor. The torque and speed characteristics are presented in Figure 9. The driving speed of the tractor rotary tillage operation is shown in Figure 10.

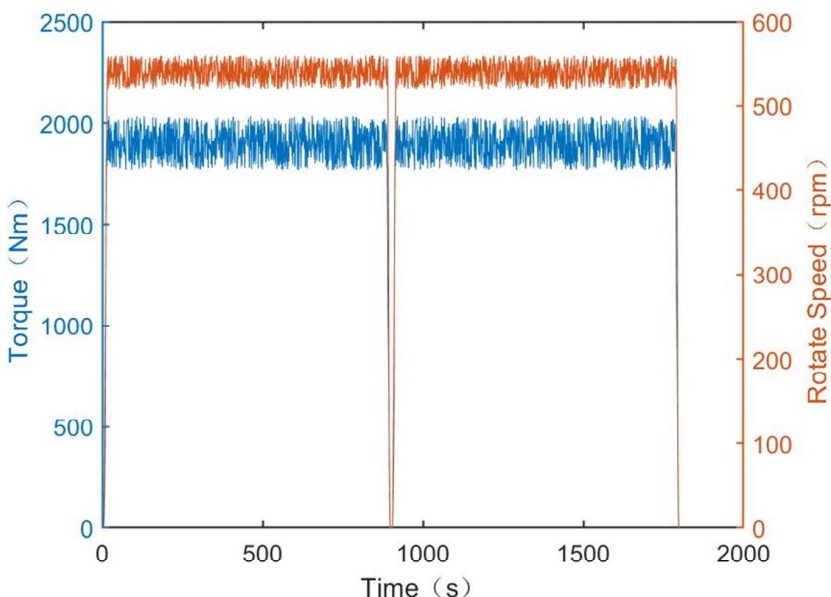

**Figure 9.** Torque and tachograph of the PTO.

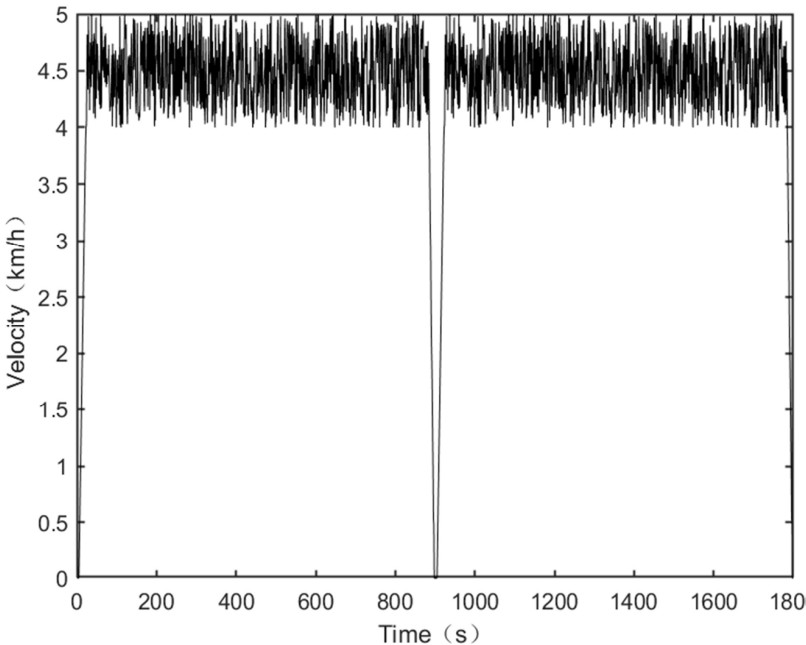

**Figure 10.** Speed of rotary tillage operations.

With respect to the tractor rotary tillage operations in the two control strategies, the motor power is shown in Figure 11; diesel power is shown in Figure 12, and battery SOC value variations are shown in Figure 13.

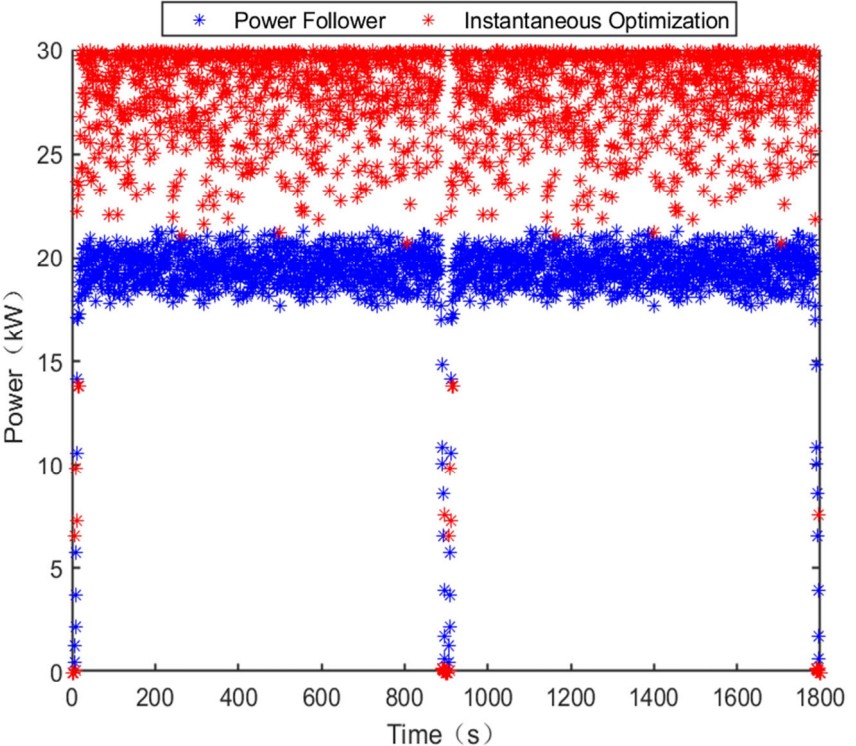

**Figure 11.** Working power of motor.

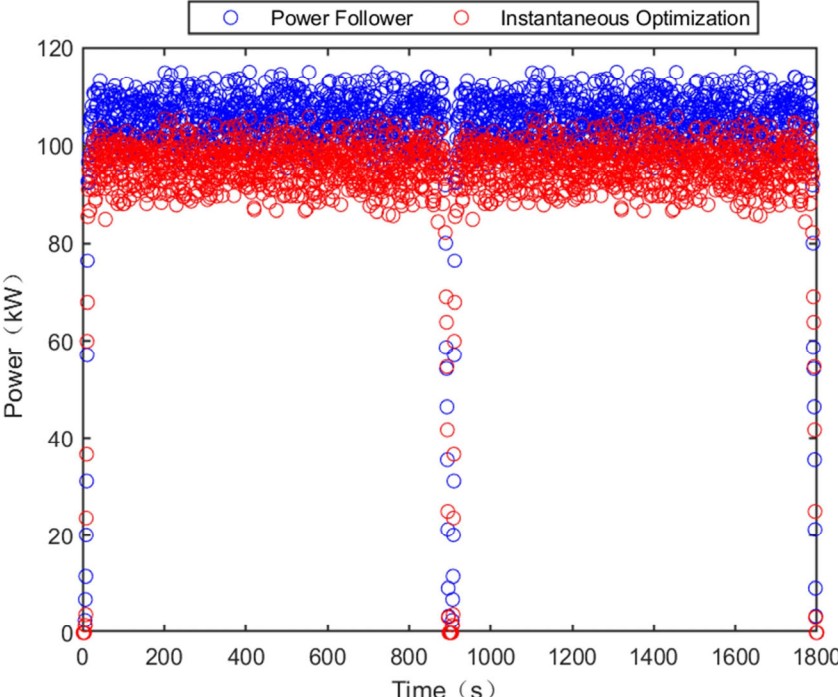

**Figure 12.** Working power of diesel engine.

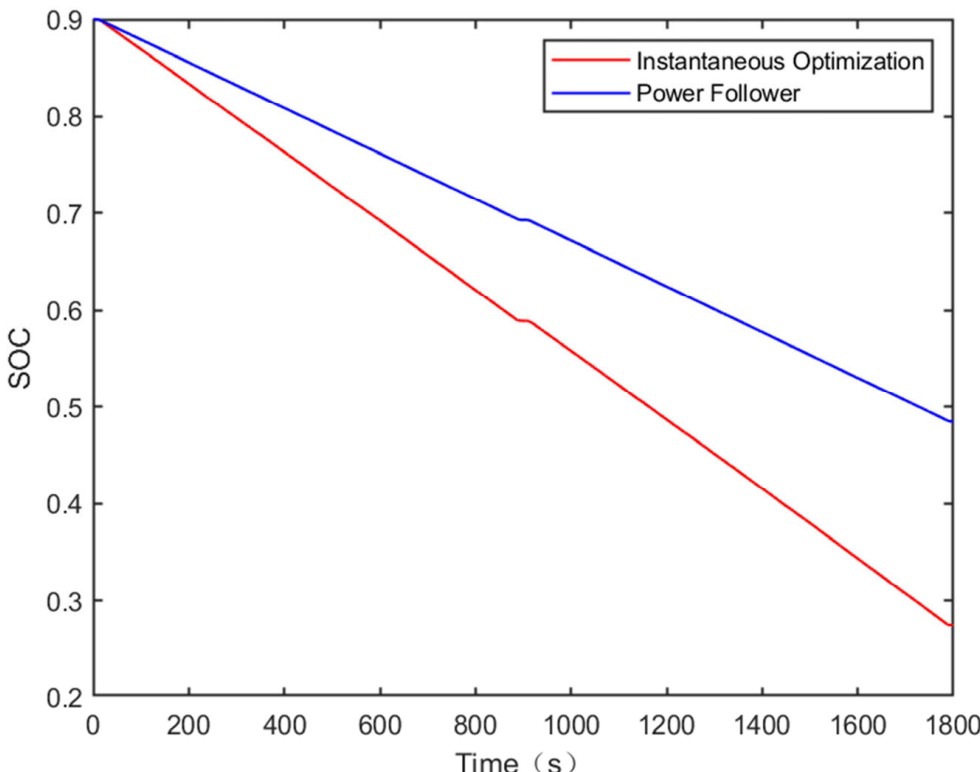

**Figure 13.** SOC state value changes.

From the simulation results, it can be seen that the load of the hybrid tractor is large, reaching 140 kW under rotary tilling conditions. Utilizing the energy-saving control strategy based on instantaneous optimization, the working power of the motor reaches more than 80% of the load. The working power of the diesel engine is mostly concentrated at 95 kW. The initial SOC value is 0.90, while the final value is 0.27. Using the power following energy-saving control strategy, the working power of the motor is relatively small and concentrated at approximately 20 kW. The maximum power of the diesel engine is approximately 115 kW, while the minimum power is approximately 100 kW. The initial SOC value is 0.90, while the final value is 0.48.

The MAP of the two control strategies for the diesel engine and motor under rotary tilling operation are presented in Figures 14 and 15, respectively.

The control strategy based on instantaneous optimization can dynamically adjust the operating regions of the motor and diesel engine according to the change of torque demand; subsequently, the diesel engine works near the optimal power curve with good dynamics and small power. The torque distribution of the energy-saving control strategy based on power following is based on the given ratio, where the adjustable range is small. The working area of the diesel engine is large, whereas that of the motor is small.

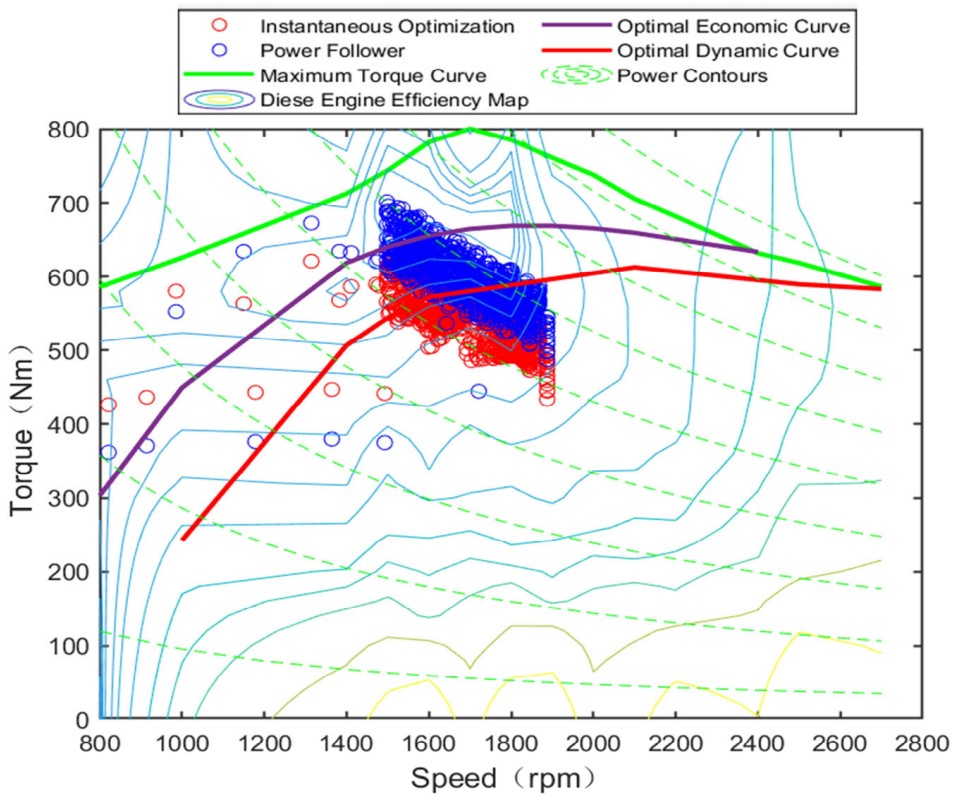

**Figure 14.** Diesel engine operating points.

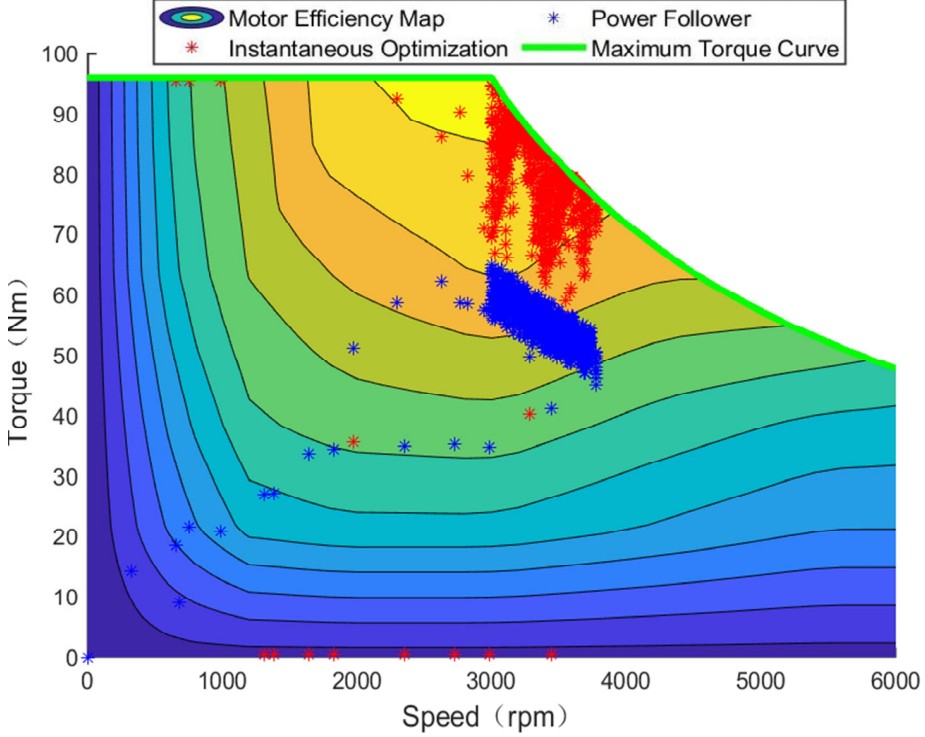

**Figure 15.** Motor operating points.

### 5.2. Analysis of Results Under Plow Condition

The tractor plowing speed is shown in Figure 16, and the plowing resistance is shown in Figure 17.

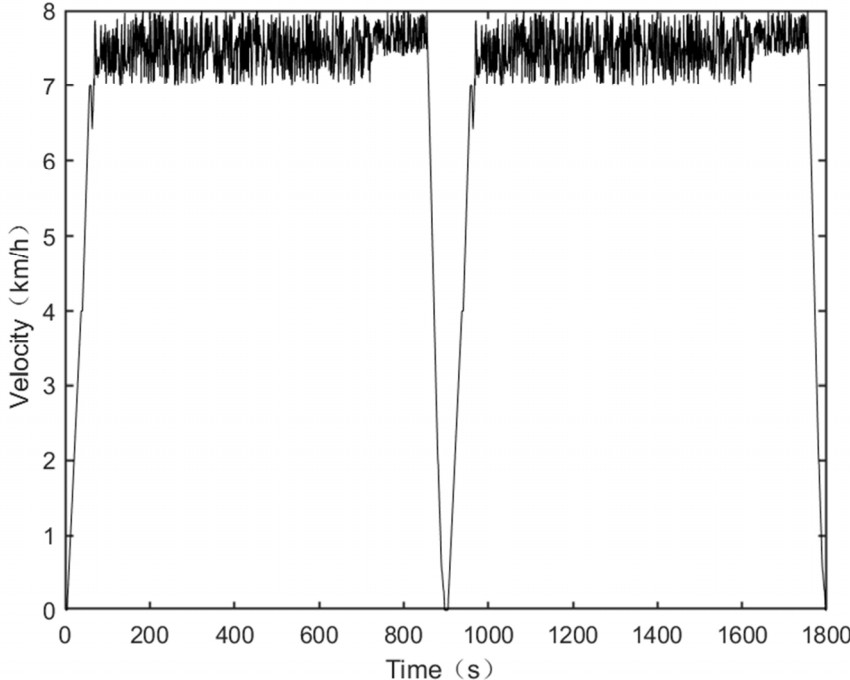

**Figure 16.** Plowing speed.

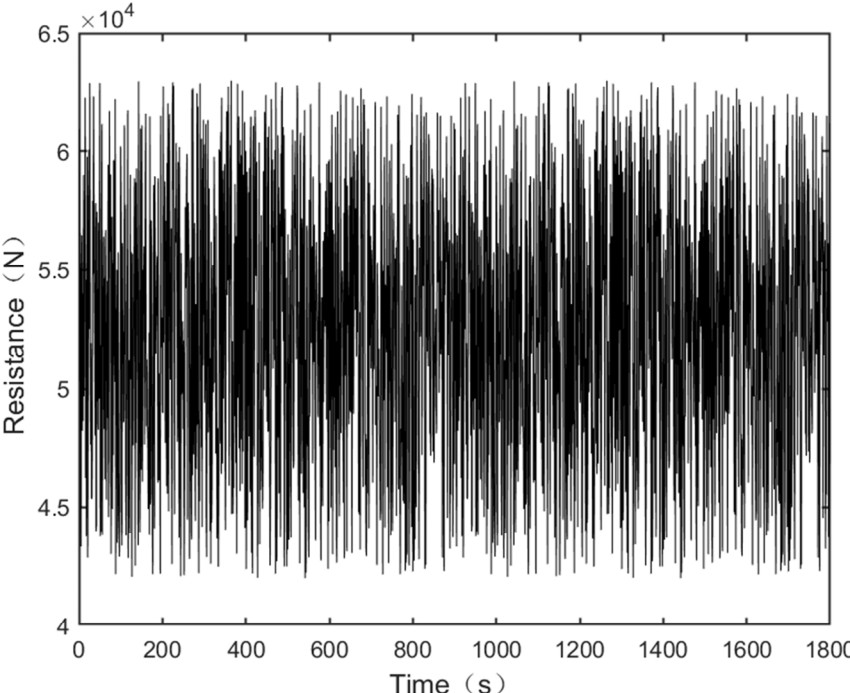

**Figure 17.** Resistance of plowing operation.

For tractor plowing operations conducted using the two control strategies, the motor power is shown in Figure 18, diesel power is shown in Figure 19, and battery SOC value variations are shown in Figure 20.

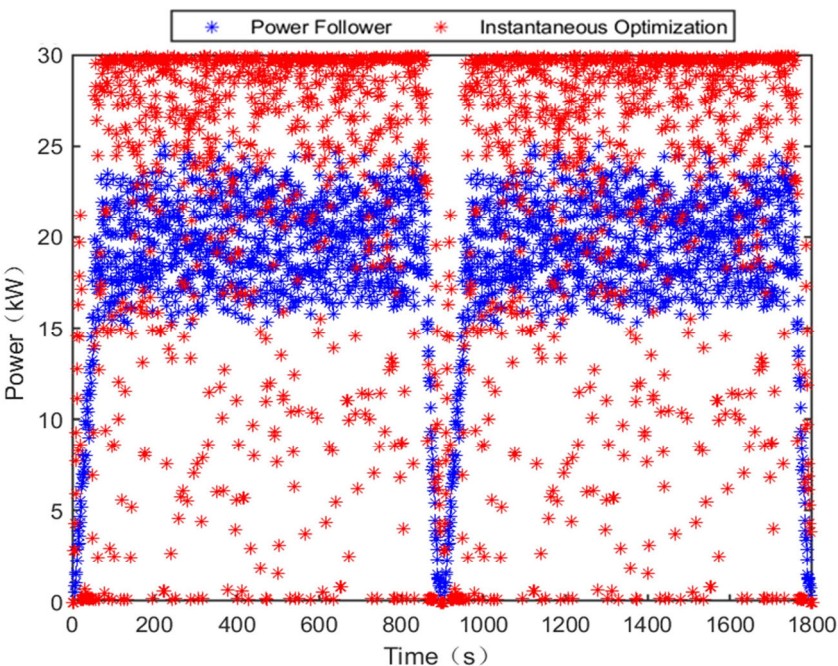

**Figure 18.** Working power of motor.

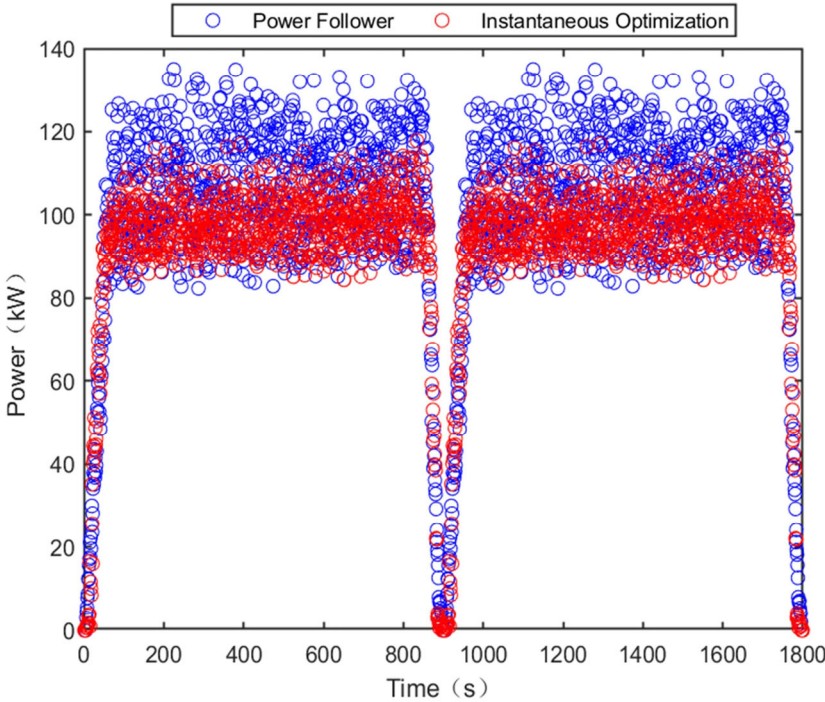

**Figure 19.** Working power of diesel engine.

It can be seen from the simulation results that under plowing conditions, based on the energy-saving control strategy of instantaneous optimization, the working power of the motor is dispersed while that of the diesel engine is concentrated. The diesel engine power is mostly between 85 and 105 kW. The initial SOC value is 0.90, while the final value is 0.42. Based on the power following the energy-saving control strategy, the working power of the motor was concentrated between 17 and 24 kW. The power of the diesel engine is large

over a significant range between 85 and 130 kW. The initial value of SOC is 0.90, while the final value is 0.50.

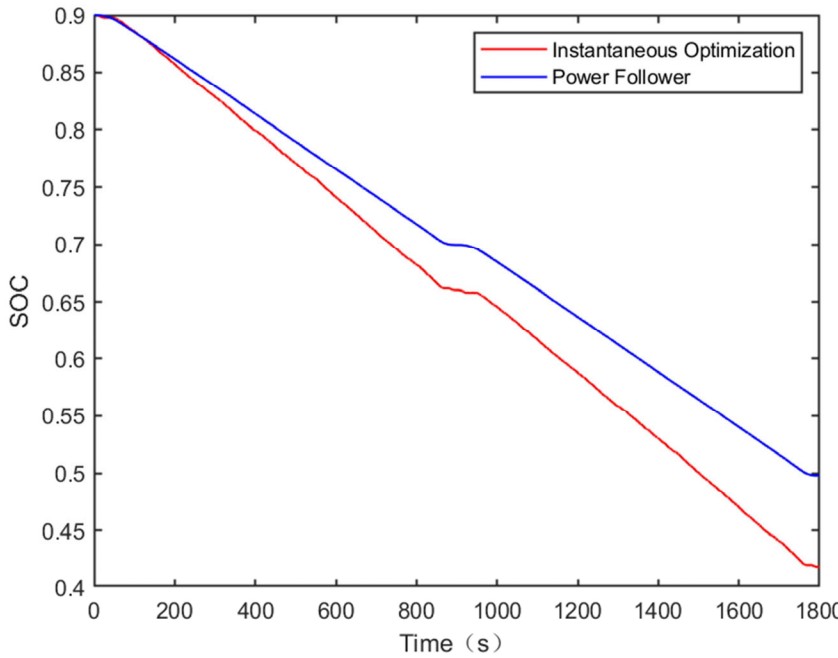

**Figure 20.** SOC state value changes.

The MAP of the diesel engine and motor for the two control strategies under plowing conditions are shown in Figures 21 and 22, respectively.

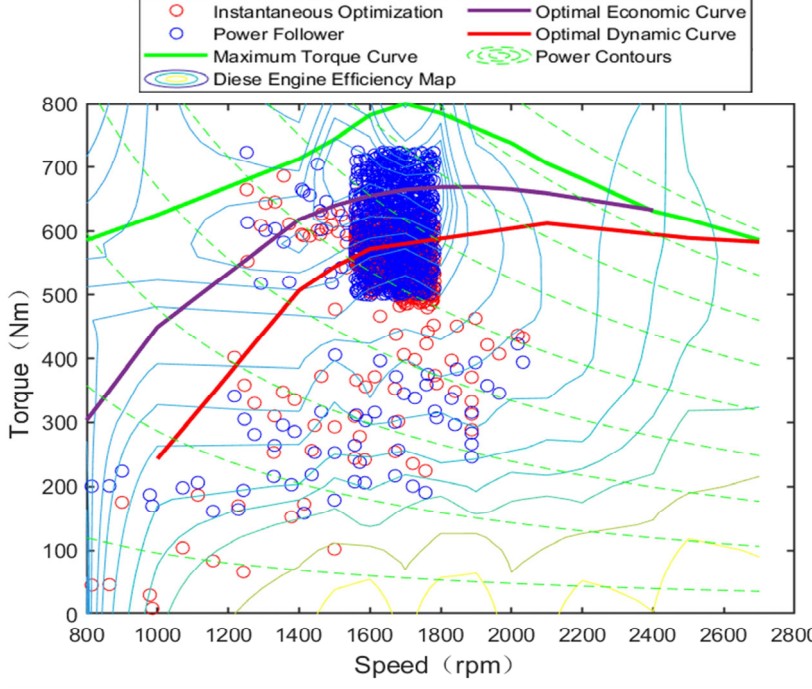

**Figure 21.** Diesel engine operating points.

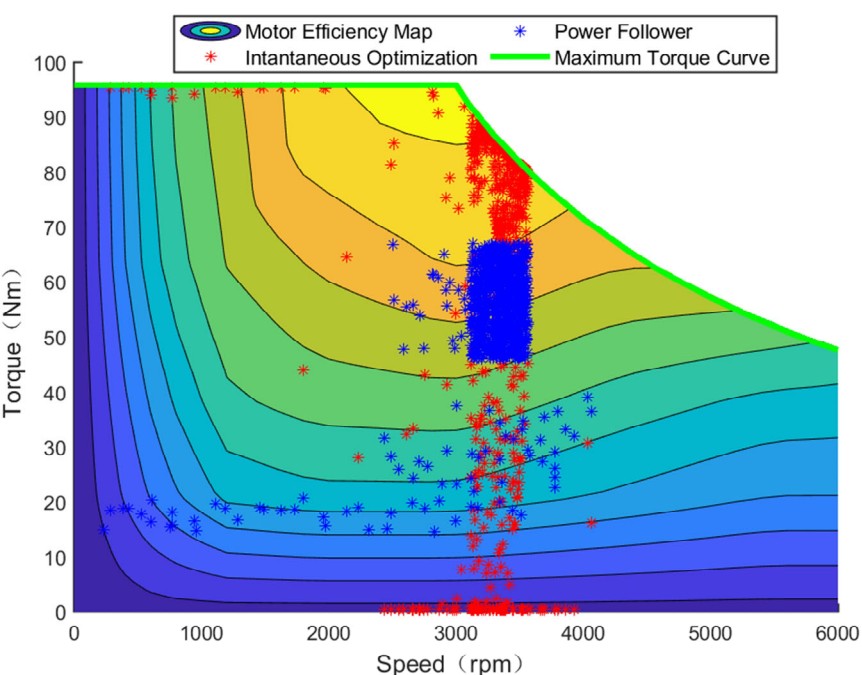

**Figure 22.** Motor operating points.

Based on the instantaneous optimization of the energy-saving control strategy, the overall working area of the diesel engine is low, near the optimal power curve. The fuel consumption is low with good dynamic performance. The efficiency within the overall operating region of the motor is also high, which improves the efficiency of the energy utilization. Based on the power following energy-saving control strategy, the working torque of the diesel engine at constant speed is large, the working power is large, and fuel consumption is increased. The overall operating region of the motor has a low efficiency and poor energy utilization.

## 6. Conclusions

This study describes an energy-saving control strategy for a hybrid tractor based on an instantaneous optimization algorithm. The objective is to minimize the equivalent fuel consumption of the entire machine. The motor torque and diesel engine torque are the control variables, while the state of charge of the power battery is the state variable. Finally, the instantaneous optimal distribution of torque is obtained.

Considering 220 hp tractors as the research object, the simulation models of the main components were built based on the topology of a parallel diesel-electric hybrid tractor. Finally, a control simulation model was built using MATLAB. To solve the problem of low energy utilization efficiency of hybrid tractors, an energy-saving control strategy based on instantaneous optimization is proposed and compared with the energy-saving control strategy based on power following. The results demonstrate that the proposed energy-saving control strategy based on instantaneous optimization effectively improves the energy efficiency of the tractor during plowing and rotary tillage and reduces equivalent fuel consumption. Under rotary tillage condition, the equivalent fuel consumption of an energy-saving control strategy based on power following is 14.46 L, whereas the equivalent fuel consumption of an energy-saving control strategy based on instantaneous optimization is 13.78 L. Accordingly, the equivalent fuel consumption decreases by 4.70%. Under plowing condition, the equivalent fuel consumption is 14.10 L based on the power following energy saving control strategy and 13.21 L based on the instantaneous optimization energy saving control strategy. Accordingly, the equivalent fuel consumption decreases by 6.31%.

The instantaneous optimal control strategy was designed for a 220 hp hybrid tractor, and acceptable results were obtained. In the future, we will design and test an instantaneous optimization control strategy framework that can meet a variety of different types of hybrid tractors. In addition, researchers can improve the energy-saving control effect of hybrid tractors by constructing models of the diesel engine and motor with more accurate dynamic simulation.

**Author Contributions:** Conceptualization, J.Z. and G.F.; methodology, L.X.; software, G.F.; validation, J.Z., G.F. and L.X.; formal analysis, M.L.; investigation, X.Y.; resources, L.X.; data curation, W.W.; writing—original draft preparation, G.F.; writing—review and editing, J.Z.; visualization, X.Y.; supervision, M.L.; funding acquisition, L.X. All authors have read and agreed to the published version of the manuscript.

**Funding:** This research was funded by National Key Research and Development Program (Grant No. 2022YFD2001203), Key Scientific and Technological Research Projects in Henan Province (Grant No. 222102240088), and the State Key Laboratory Open Project (Grant No. SKT202200).

**Data Availability Statement:** Not applicable.

**Conflicts of Interest:** Wei Wang and Mengnan Liu are employees of YTO Group Corporation, Luoyang 471004, China. The paper reflects the views of the scientists, and not the company.

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
