# Peer review of "Energy-Saving Control of Hybrid Tractors Based on Instantaneous Optimization"

_wevj, doi:10.3390/wevj14020027_

Round 1

Reviewer 1 Report

The aim of the paper is to minimize the fuel consumption of a hybrid tractor by simulating an energy saving control strategy based on instantaneous optimization of the required torque; this strategy allows a fuel consumption reduction in respect of the power following strategy. The simulations regard the plowing and rotary tillage working conditions for a 220 hp tractor.

The limitations of the work are not addressed, in particular in the Conclusion section: (for example) why do the authors use only one model of tractor to test the effectiveness of their methodology?

Moreover, there is the need for additional information about numerical model and optimization procedures. Furthermore, there are no information about the tractor model used for the simulation, nor information about the measurement of power absorption for the rotary tillage and the plowing operations.

The writing of the equations should be improved.

Methodology

Further details should be provided about the numerical motor model Map diagram, as long as the authors did for the diesel engine.

Section 4.1.2 step 4 - Is it “J” Qc? Please clarify to better understand the optimization procedure.

Typing mistakes or omissions

Table 2 lacks the unit measure of speed km/h

Line 161 – edit “Pdirve

Line 161-168 check the parameters description please. α parameter description lacks. By the equation 5 m seems to be the mass of the tractor, while α the tilt of the ground.

Line 181 – Edit the subscript of FTN driving force

Line 290 – edit instsntaneous

Figure 6 –The calculation of Te and Tm seems to be in contrast with equation (1) please check

Figure 15 – Edit the Chart legend

Reviewer 2 Report

The manuscript details an energy-saving control strategy for a hybrid tractor (diesel-electric + battery) based on instantaneous optimization. Through instantaneous optimization, the engine is allowed to operate more on optimal torque and efficiency, thus reducing the fuel consumption by about 4-6% compared to the power following method. The results constitute an incremental improvement in control strategies and should be of interest to the readers of World Electric Vehicle Journal. 

I have only some minor comments to the manuscript: 
- For the battery, the SOC is reduced from 0.9 to 0.25-0.5 in 30 minutes (1800 s). In my understanding, the battery through the control strategy, is what helps reduce the fuel consumption, but this would only last for about 1 hr until it is depleted, and any work beyond this would not have any saving since there the diesel engine have to supply the full power demand. Did the authors consider having recharging of the battery during some phases as part of the control strategy? Wouldn't it be more suitable to scale and design the battery to last one working day or at least 4-5 hrs? 

- The proposed instantaneous optimization model is only compared to a power following model and the main argument and motivation is the fuel savings compared to this model. Why are not other strategies considered such as a rule-based management strategy or a global optimization (not instantaneous) control strategy? This would further strengthen the argument proposed since the energy management strategy in itself is not new. 

Round 2

Reviewer 1 Report

The conclusion section is still not covering limitations of the work. Why do the authors not integrate that Section with the contents of responses to Point 1 and Point 2? The limitations of the work are very important to better understand the research study domain and its future potential improvements.

Author Response

请参阅附件。
